# Longitudinal Investigation of the Native Grass Hay from Storage to Market Reveals Mycotoxin-Associated Fungi

**DOI:** 10.3390/microorganisms10061154

**Published:** 2022-06-02

**Authors:** Shuai Du, Sihan You, Xiaowei Jiang, Yuyu Li, Yushan Jia

**Affiliations:** 1National Engineering Laboratory of Biological Feed Safety and Pollution Prevention and Control, Key Laboratory of Molecular Nutrition, Ministry of Education, Key Laboratory of Animal Nutrition and Feed, Ministry of Agriculture and Rural Affairs, Key Laboratory of Animal Nutrition and Feed Science of Zhejiang Province, Institute of Feed Science, Zhejiang University, Hangzhou 310058, China; nmgdushuai@zju.edu.cn; 2Key Laboratory of Forage Cultivation, Processing and High Efficient Utilization, Ministry of Agriculture, Key Laboratory of Grassland Resources, Ministry of Education, College of Grassland, Resources and Environment, Inner Mongolia Agricultural University, Hohhot 010019, China; you_sihan@163.com (S.Y.); lyy2017@emails.imau.edu.cn (Y.L.); 3Institute of Grassland Research, Chinese Academy of Agricultural Sciences, Hohhot 010020, China; jxwayy@126.com

**Keywords:** native grass hay, mycotoxin, fungal diversity, storage period

## Abstract

This study aimed to characterize the fungal diversity and mycotoxin concentrations of native grass hay in various storage periods. In the present study, the native grass hay samples were collected when stored for 0 d (D0 group), 30 d (D30 group), and 150 d (D150 group), respectively. Here, mycotoxin analyses combined with ITS gene sequence were performed to reveal the changes in response to the storage period. There were notable differences in deoxynivalenol and aflatoxin B_1_ concentrations among the three groups. Compared to the D150 group, the diversity of the fungal community was higher in the D0 and D30 groups, which indicating the diversity was significantly influenced by the storage period. No significant (*p* > 0.05) difference was observed among the three groups on the dominant phyla. Interestingly, a significant (*p* < 0.05) difference was also observed in *Chactomella* and *Aspergillus* among the three groups, the abundance of the *Chactomella* was significantly (*p* < 0.05) decreased and the abundance of *Aspergillus* was statistically (*p* < 0.05) increased in the D150 group. Correlation analysis of the association of fungi with mycotoxin could provide a comprehensive understanding of the structure and function of the fungal community. These results indicated that the good practices of storage are essential for the prevention of mycotoxin. The information contained in the present study is vital for the further development of strategies for hay storage with high quality in the harsh Mongolian Plateau ecosystem.

## 1. Introduction

Mycotoxins are natural chemical contaminants and secondary metabolites produced by fungi, which are toxic to animals with high-risk factors and result in a significant threat to human health [1]. Mycotoxin formation is directly associated with many biological factors, such as harvesting, processing conditions and storage period [1,2]. This study was focused on the well-known mycotoxins, including aflatoxins B_1_ (AFB_1_), deoxynivalenol (DON), zearalenone (ZEN), and ochratoxin A (OTA), in hays [1,3].

Native grassland is widely distributed in the Mongolian Plateau and could provide sufficient nutrition for ruminants. Traditionally, grazing was the most common feeding system, but in recent years, the native grass hay (bales) have partially or totally replaced grazing for the ruminants in the Mongolian Plateau [4]. Ensiling is a traditional way of animal feed and green forage preservation because it can provide green forages for animals year-round, prolonging the storage time and effectively decreasing the nutrition lost in forages [5]. However, the moisture content, the water-soluble carbohydrate content, and the numbers of epiphytic lactic acid bacteria microflora were lower than the requirement, making it difficult to produce silage with high quality [5]. Native grass hay has become the main feeding system in Inner Mongolia Plateau [4].

Hay, which is stored in an aerobic, dry, and neutral pH environment, can be easier contaminated by fungi compared to silage [3]. Additionally, some fungal species can produce exorbitant amounts of fungal spores that become airborne when dried hay bales are moved and broken up [3]. Previous reports have found that hay has been shown to have high levels of *Aspergillus* and *Penicillium*, as well as mycotoxins [6,7], despite the hay having dried in the field and having been stored under dry conditions [8,9]. Mycotoxins in silage for ruminants have been studied extensively [7,10,11,12], though less information is known connected with the fungal community and their metabolites in hay [3,6,9,13], especially in native grass hay.

Studies on forage derived from grass have so far focused on the presence of either specific bacterial species and metabolites production, and not the overall fungal community and mycotoxins. Therefore, the present aimed to characterize the fungal diversity and some mycotoxins of native grass hay in various storage periods.

## 2. Materials and Methods

### 2.1. Preparation of Native Grass Hay

In the present study, the native grass hay was taken from a typical steppe in Balin Left Banner, Inner Mongolian Plateau, which is comprised of *Stipa gigantea* Link.; *Leymus chinensis* (Trin.) Tzvel.; *Cleistogenes squarrosa* (Trin.) Keng, *Lespedeza daurica* (Laxm.) Schindl.; *Agropyron mongolicum* Keng.; *Allium ramosum* Link.; *Melissitus ruthenicus* (L.) Peschkova) *Erodium stephanianum* Willd.; and *Artemisia sieversiana* Ehrhart ex Willd.; *Stipa gigantea* Link.; and *Leymus chinensis* (Trin.) Tzvel. as the predominant species. The chemical compositions of the native grass was analyzed before harvesting and the native grass was mowed at the milk stage to get high-quality and quantity hay on 20 August 2018. For hay making, the native grass was harvested with stubble height of 2–5 cm by a tractor-mounted lawn mower. After harvesting, the native grass was tedded twice daily and dried for approximately 72 h under natural conditions. According to previous reports, hays with 16% or lower moisture at baling can be stored requiring little heat, with a low microbial activity and absence of mold [14,15]. To minimize the effects of moisture, at the moisture content of the dried native grass was lower than 14% the native grass hay was baled. The native grass hay was raked back into windrows and formed into square bales (75.0 × 50.0 × 40.0 cm, 20.0 ± 3.0 kg) when the grass was dried.

### 2.2. Sample Collection and Storage

The sampling methods adopted in the current study was similar the method of Ceniti et al. [12]. Briefly, each sample of about 1000 g was obtained by sampling of 27 bales randomly chosen by a motorized corer (DeWalt Wi128604, Stanley Black & Decker Co., Ltd., New Britain, CT, USA) with the length 60.0 cm and internal diameter 22 mm. The sampling devices were thoroughly sterilized before sampling a new lot of hay. Six replicates in each storage period and a total of 18 samples were used in this study. After collection, the samples were transported to the laboratory and were carefully mixed and separated in two aliquots of 600 and 400 g for mycotoxin and ITS gene sequencing, respectively. The samples were immediately frozen in liquid nitrogen and stored at −80 °C for further analysis. In the present study, the native grass hay was stored under the natural conditions in the shed in Balin Left Banner, Inner Mongolian Plateau. The shed was comprised with color steel plate that could against the rainfall, snow, and wind for the native grass hay.

The 2 g ground samples were transferred into a 50 mL polypropylene centrifuge tube and extracted with 20 mL 0.1 mL/100 mL formic acid for 5 min at 300 rpm on an orbital shaker. Then, centrifugation at 10,000 rpm for 5 min and the supernatant was collected. At last, the supernatant was dried under nitrogen at 40 °C, and dissolved with the mixture of 90 mL 0.1 mL/100 mL and 10 mL acetonitrile for analysis.

### 2.3. Mycotoxin Analyses

The ultra-high-performance liquid chromatography–tandem mass spectrometry system (Acquity I-Class, Waters, Milford, MA, USA) combined with the triple quadrupole mass spectrometer (XEVO TQ-S, Waters, Manchester, UK) was used to determine the 4 mycotoxins, including aflatoxins B_1_ (AFB_1_), deoxynivalenol (DON), zearalenone (ZEN), and ochratoxin A (OTA). The chromatographic resolution was obtained through an Agilent ZORBAX Eclipse Plus C18 analytical column (50 mm × 2.1 mm, 1.8 µm), with a velocity of flow of 5 µL/s, with a sampling volume of 10 µL. The mobile phase consisted of A (H_2_O, 0.1 mL/100 mL formic acid and 5 mol/L ammonium acetate,) and B (0.1 mL/100 mL formic acid and 0.1 mL/100 mL acetonitrile) and the column temperature was 33 °C. The multiple reaction monitoring modes (MRM) at positive polarity was used to identify ESI-MS/MS according to the method of Kafouris et al. [1]. Once the extraction procedure has been optimized, the analytical methods was validated in order to show the applicability and robustness. Once the extraction procedure has been optimized, the negative control chromatogram and ion chromatography were obtained according to the method, and the signal intensity and baseline noise were calculated. The analytical method was validated in order to show the applicability and robustness. According to the previous reports, the signal to noise ≥3 as the limit of detection and signal to noise ≥10 as the limit of quantification [16].

### 2.4. DNA Extraction and PCR Amplification

The fungal community genomic DNA was extracted with the DNA kit (Omega Bio-Tek, Norcross, GA, USA) according to the protocols from the 18 samples. The content and purity of the extracted DNA was determined by the NanoDrop 2000 UV-vis spectrophotometer (Thermo Scientific, Wilmington, DE, USA) and the quality of the extracted DNA was assessed with 1 g/100 mL agarose gel electrophoresis. The sequences for the ITS1 and ITS2 regions of the fungal ITS gene were amplified with forward and reverse primers as follows: ITS1F (5′-CTTGGTCATTTAGAGGAAGTAA-3′) and ITS2R (5′-GCTGCGTTCTTCATCGATGC-3′). The polymerase chain reaction amplification was conducted by Majorbio Biopharm Technology Co., Ltd. (Shanghai, China). The ITS1 portion of the rRNA gene was described at the MiSeq platform (Shanghai Majorbio Biopharm Technology Co., Ltd.).

### 2.5. Bioinformatics and Statistical Analysis

Raw fastq files were subjected to quality control by FLASH. The operational taxonomic units (OTUs) were clustered with a 97% similarity through UPARSE. The common and unique OTUs was used to construct the Venn diagram by R (version 1.6.2). The alpha diversity for these samples were used the Chao1 value and Shannon index to evaluate richness and diversity [17,18,19,20], and Good’s coverage were calculated through QIIME software. Principal coordinates analysis (PCoA) was conducted on OTU level by R (version 3.3.1) software based on weighted UniFrac distances. The taxonomy was according to ITS gene sequence through the ITS database (unite8.0/its_fungi) at a confidence threshold of 70%. The main differentially abundant genera were analyzed by the linear discrimination analysis coupled with effect size (LEfSe) method. The Fungi Functional Guild (FUNGuild) tool (http://www.funguild.org/ (accessed on 17 April 2022)) were used to predict the genes’ metabolic pathways. The Kyoto Encyclopedia of Genes and Genomes (KEGG) Module database were also used to assign the genes into enzyme functions.

The mycotoxins and alpha diversity were analyzed by SAS version 9.0 software. Statistical differences among means were determined through analysis of variance (ANOVA). The significant differences were considered at *p* < 0.05 level.

## 3. Results

There were notable differences in mycotoxin contamination during the storage period (Figure 1). Significant (*p* < 0.05) differences were observed in DON among the three groups, in the order of D30 (27.69 µg/kg), D150 (20.64 µg/kg), and D0 groups (14.11 µg/kg). Compared to the D0 (6.71 µg/kg) and D30 (6.69 µg/kg) groups, the AFB1 concentration markedly (*p* < 0.05) increased with days of storage and especially in the D150 (7.82 µg/kg) group. Interestingly, there was no significant (*p* > 0.05) difference in OTA among the three groups. Significantly (*p* < 0.05) higher levels of ZEN concentration were found in the D30 (5.53 µg/kg) group compared to that in the D0 (3.27 µg/kg) and D150 (3.21 µg/kg) groups, while no significant (*p* > 0.05) difference was found between the D0 and D150 groups.

A total of 1,131,633 high-quality sequences and 1,001 operational taxonomic unit (OTUs) numbers at 97% identity were collected by ITS Gene Amplification from 18 samples. The Good’s coverage index of all samples was more than 99% (data not shown). According to the Shannon index, there were no significant (*p* > 0.05) difference in diversity among the three groups (Figure 2A), while a significant (*p* < 0.05) difference was observed in microbiota richness among the three groups, indicating less richness in the D150 group (Figure 2B). In addition, the Venn plot shows that the groups shared 341 OTUs, while D0, D30 and D150 groups had 197, 150, and 81 exclusive OTUs, respectively (Figure 2C). To characterize the effects of storage time on beta diversity, the weighted UniFrac distance was used to address the fungal community in all samples (Figure 2D). The PCoA profile of fungal community on OUT level displayed that the composition of the fungal community was distinctly separated at each time point from the three groups.

Taxonomic analysis indicated the presence of 54 genera belonging to 9 phyla, whereas, on the phylum level, only 2 phyla were referred to as the detected phyla (the abundance was higher than 1% at least in one group), with *Ascomycota* and *Basidiomycota* being the predominant phyla accounting for higher than 95% of the total reads, and no significant (*p* > 0.05) difference was found in these phyla among the three groups (Figure 3A). On the genus level, 20 genera were considered as the detectable genera (the abundance was higher than 1% at least in one group). The main genera included *Chactomella*, *Aspergillus*, *Alternaria* and *Didymella* (Figure 3B). Additionally, a significant (*p* < 0.05) difference was also observed in *Chactomella* and *Aspergillus* among the three groups (Figure 3C).

Furthermore, the LEfSe results illustrated that there was a significant difference in fungal community in different storage time points from the three groups (Figure 4A,B). Compared to the D0 and D30 groups, the abundance of the *Chactomella* was significantly (*p* < 0.05) decreased and the abundance of *Aspergillus* was significantly (*p* < 0.05) increased in the D150 group, respectively.

The ITS gene-predicted functional profile of the fungal community is presented in Figure 5. Fungal communities were analyzed by the FUNGuild online tool. As shown in Figure 5A, more than six fungal function groups were inferred by FUNGuild. After storage, the main fungal functional group was Saprotroph, and it continued to the storage of 150 d. In addition, the fungal functional groups inferred by FUNGuild, indicating that the storage decreased the animal pathogen and plant pathogen; the storage could inhibit these pathogenic microbes and improve the safety and quality of native grass hay. The functional predictions based on the KEGG Module database indicated that the relative abundances exhibited several changes in key enzymes (Figure 5B). The abundance of the top 30 key enzymes were decreased in the D150 group compared to that in the D0 group.

Pearson’s correlation analysis was conducted to further identify potential correlations between changes in fungal microbiota and mycotoxin (Figure 6). The results of the present study showed the genus *Aspergillus* was statistically (*p* < 0.001) positively associated with AFB_1_.

## 4. Discussion

This study characterized the changes in mycotoxin and fungal community structure of native grass hay in response to storage period by mycotoxin parameters analyses combined with ITS gene sequences, and has provided new insights into the effects of storage period on native grass hay.

Mycotoxin contamination is a worldwide problem for various agricultural and husbandry commodities, including both pre- and post-harvest and storage [7]. As expected, the most studied are regulated AFB_1_, DON, OTA, ZEN and others [7]. A previously published study indicated that the weather conditions could directly affect mold spoilage, and favorable environmental conditions are critical for the formation of mycotoxin [12]. In the present study, lower concentrations of DON and ZEN were found in the D0 group, and higher concentrations of DON and ZEN were observed in the D30 group (Figure 1), which could be due to the contribution of the weather conditions in late autumn, which could stimulate the development of fungi and lead to the formation of DON and ZEN [19]. Furthermore, the concentration of AFB_1_ markedly increased in the D150 group, which could be the contribution of the fungi capable of producing this mycotoxin being detected at a high level (Figure 3B).

In the current study, the storage period directly influenced the fungal community compositions of native grass hay. Compared to the D30 and D150 groups, the higher numbers of OTUs and Chao1 index were detected in the D0 group. Significant differences were observed in the Shannon index, indicating that the D0 group had the higher community evenness than that in the D30 and D150 groups. The changes in the fungal community structure were also explored, these results illustrated that the three groups distinctly separated from different time point, as reflected by the clustering of the samples by storage period using PCoA. Macroscopically, the storage period drove a separation in the fungal community (Figure 2D), the distinguishable changes among the three groups, following the reports that noticeable separation of the fungal community structure was observed in silage and hay during the various storage periods [3,20,21]. Various treatments and storage of hay favor different fungal contaminants and the storage period is the key factor in determining microbial community structure may be the main reason [3].

The phylum-level core microbiomes were *Ascomycota*, accounting for about 80% of the total fungal species [12], which is in agreement with the report that the genus Ascomycete represents a range of fungi that are commonly connected with plant material [22]. At the genus level, *Chaetomella*, *Aspergillus*, *Alternaria* and *Didymella* were the dominant genera, which is similar to the previous study that *Aspergillus* and *Alternaria* were the primary genera in hay [12]. The genus *Chaetomella* as plant pathogens and saprotroph are widely distributed in both temperate and tropical regions that growing on soil [23,24]. In the current study, the abundance of *Chaetomella* was decreasing with the increase of storage period, which could be caused by the fact that the native grass was dried on the grassland until the moisture was lower than 14% and the genus *Chaetomella* was detected with higher abundance in the D0 group. Similarly, the relative abundance of plant pathogens, fungal parasites, plant pathogens, plant saprotrophs, and plant pathogen-undefined saprotrophs also dropped with the decrease of *Chaetomella*. One previously published study indicated that different treatments and storage for hay lead to various fungal contaminants, and are more likely to be contaminated with some fungal species that could tolerate the lower water activities, including *Aspergillus* spp. [3]. The presence of *Aspergillus* has been reported in silage, haylage and hay, and the most important mycotoxins, Afs, are produced by these organisms [3,7,12]. In the present study, the relative abundance of *Aspergillus* was significantly increased in the D150 group, which indicated the long storage period could more likely be contaminated by *Aspergillus*. There result is similar with the previous report that large amounts of *Aspergillus* were found in oat [25], which could be explained by some unique characteristics of the *Poaceae*. *Alternaria* is a ubiquitous and saprophytic fungus that commonly exists in dead plant materials and is also a plant pathogen causing disease in several crops [26,27], and relates to the production of toxins, including alternariol, altenuene, tenuazonic acid, and altertoxin [28]. In the current study, the native grass hay was stored from August 2018 to January 2019. The temperature and relative humidity of the storage environment varied. Additionally, prior research found that diverse conditions during the storage period benefit the growth of microorganisms [29]. Therefore, the abundance of Alternaria was also increased with the increase of the storage period.

Various fungi were correlated with different metabolic pathways, revealing that multiple metabolic pathways were active during the different storage periods. These metabolic functions allowed the fungi to grow, proliferate and respond to the environment [30]. Various fungi were connected with mycotoxins, indicating the mycotoxins were actived by fungal microbiota. In the present study, the significantly higher AFB_1_ concentration in the D150 group, which could be explained by the higher abundance of *Aspergillus* in the D150 group. This result indirectly explained an observation of the correlation between fungal microbiota and mycotoxin. DON and ZEN are widely found in foods and feeds. *Fusarium* species, including *Fusarium graminearum*, *Fusarium tricinctum*, *Fusarium culmorum*, *Fusarium equiseti*, *Fusarium sernitectum*, and *Fusarium solani*, are the primary strains that could directly produce the DON and ZEN toxins [31,32], and the animals’ health will be seriously endangered once DON-contaminated feed is consumed [33]. However, in the present, the abundance of *Fusarium* was not observed, because it was lower than 1‰ (data not shown).

## 5. Conclusions

This study explored the mycotoxins and fungal community of native grass hay during the various storage periods. These results indicated that the good practices of storage are essential for the prevention of mycotoxins. One of the limitations of the current study was the lack of chemical compositions. The information obtained in this study is vital for the further development of strategies for hay storage with high quality in the harsh Mongolian Plateau ecosystem.

## Figures and Tables

**Figure 1 microorganisms-10-01154-f001:**
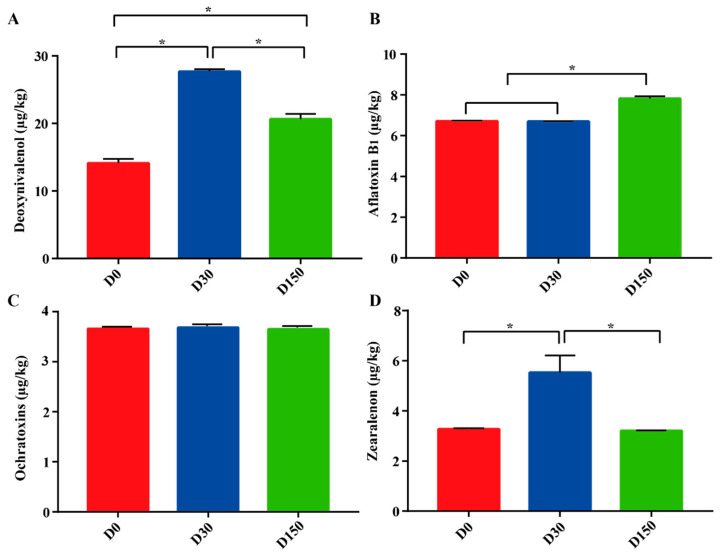
Mycotoxins across the three storage times (n = 6). (**A**) Deoxynivalenol. (**B**) Aflatoxin B1. (**C**) Ochratoxins A. (**D**) Zearalenone. D0, storage for 0 d; D30, storage for 30 d; D150, storage for 150 d. * Indicates significant difference among the three groups at *p* < 0.05 level.

**Figure 2 microorganisms-10-01154-f002:**
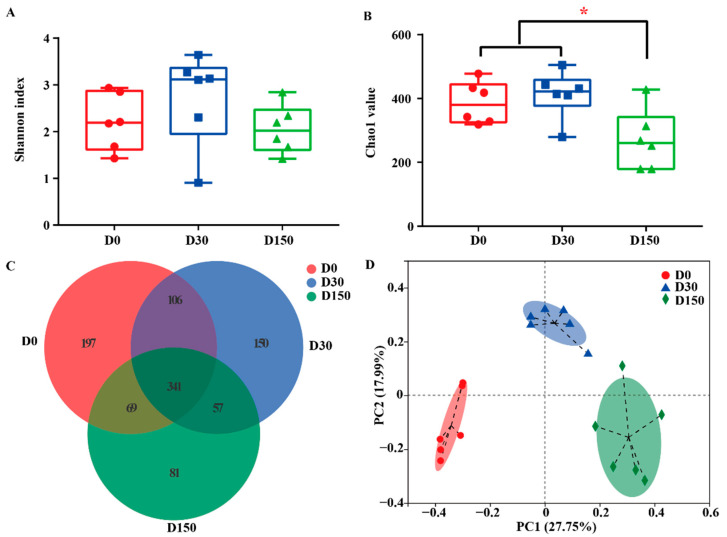
Fungal diversity across the three storage times (n = 6). (**A**) Fungal diversity was estimated by the Shannon index. (**B**) Fungal richness is estimated by the Chao1 value. (**C**) Venn diagram representing the common and unique OTUs found at each storage time point from the three groups. (**D**) Principal coordinate analysis (PCoA) of samples was conducted based on weighted UniFrac distance from the three groups. D0, storage for 0 d; D30, storage for 30 d; D150, storage for 150 d. * Indicates significant difference among the three groups at *p* < 0.05 level.

**Figure 3 microorganisms-10-01154-f003:**
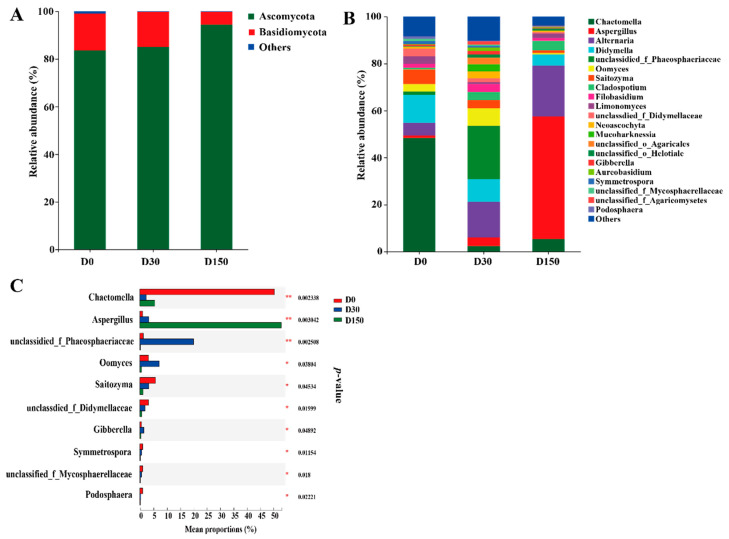
Fungal community structure across the three storage times (n = 6). (**A**) Phylum-level compositions of the fungal community in the three storage times. (**B**) Genus-level compositions of the fungal community in the three storage times. (**C**) The extended error bar plot displaying the significant differences among storage periods at the genus level. D0, storage at 0 d; D30, storage 30 d; D150, storage 150 d.

**Figure 4 microorganisms-10-01154-f004:**
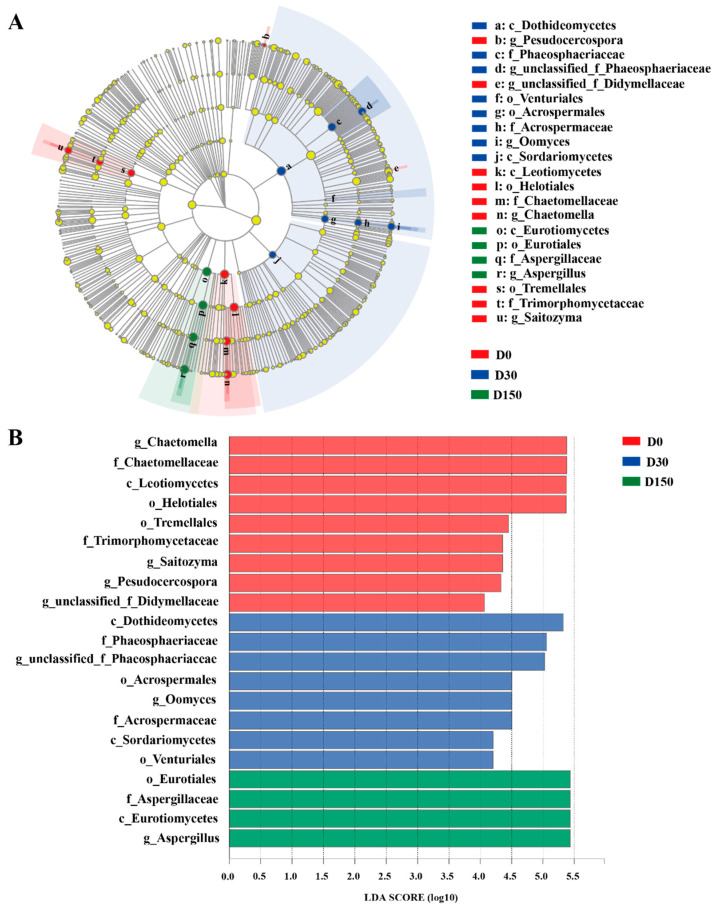
The cladogram (**A**) and histogram (**B**) show fungal taxa with a linear discriminant analysis (LDA) score > 4.0 using linear discrimination analysis coupled with effect size analysis.

**Figure 5 microorganisms-10-01154-f005:**
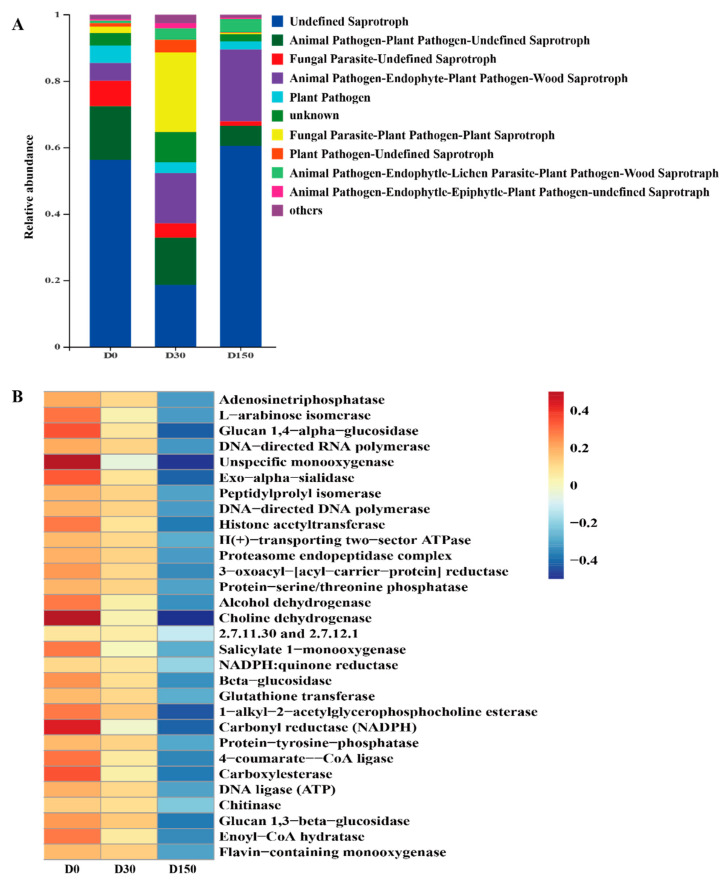
ITS gene-predicted fungal functional profiles during storage analyzed by FUNGuild and PICRUSt (n = 6). (**A**) Variations in the composition of fungal functional groups inferred by FUNGuild. (**B**) Level 3 KEGG ortholog functional predictions of the relative abundances of the top 30 enzymes.

**Figure 6 microorganisms-10-01154-f006:**
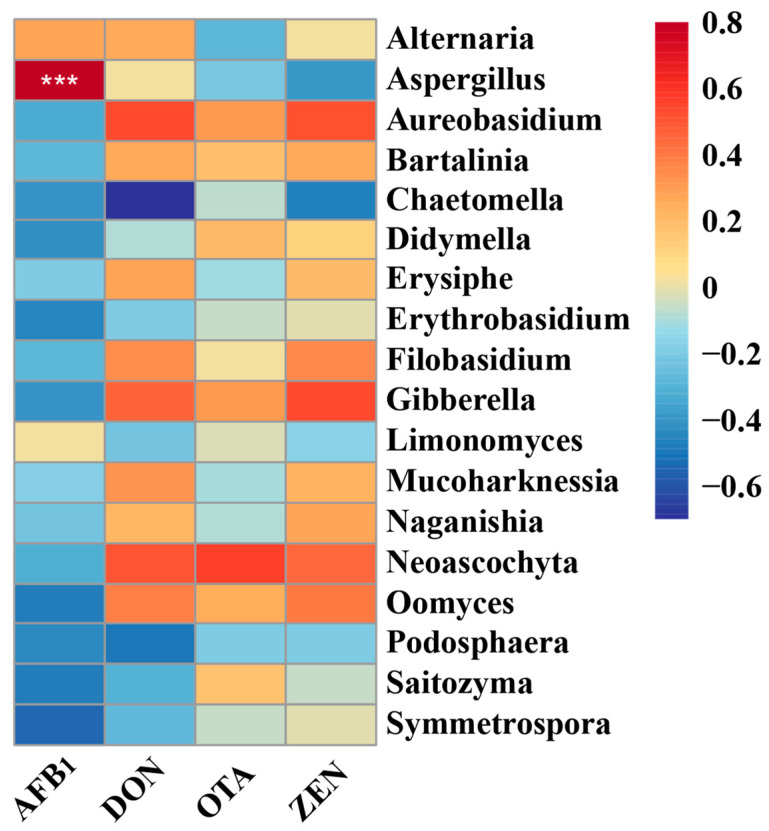
Heatmaps of Pearson’s correlations between dominant genera and mycotoxin. Red represents a positive correlation, while blue represents a negative correlation. Levels of significant are shown as follows: *** *p* < 0.001. AFB1, Aflatoxin B1; DON, Deoxynivalenol; OTA, Ochratoxins A; ZEN, Zearalenone.

## Data Availability

The raw reads of the 16S rRNA sequencing were uploaded to the Sequence Read Archive with accession number PRJNA844344.

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
