# Peer review of "Longitudinal Investigation of the Native Grass Hay from Storage to Market Reveals Mycotoxin-Associated Fungi"

_microorganisms, 2022, doi:10.3390/microorganisms10061154_

Round 1

Reviewer 1 Report

The authors draw relevant data for a significant source of animal feed which is hay, the relationship among toxin production and specific type of fungal genera is explored. Also, there is important information regarding the description of the microbiota present in hay during storage. However, the author may want to consider revising some details in their manuscript.     

Abstract

Line 16 And mycotoxin what? Levels? Frequency? Diversity?

Line 17 were collected and stored?

Introduction

In general introduction is quite ok, it could maybe benefit from some restructuring of some ideas for order and rephrasing of some ideas that are somewhat difficult to follow. Please check composition of sentences and haphazard ideas.    

Line 43 “but in recent years wrapped in bales (hay)” please rephrase

Materials and methods

Lines 85 and 86 “(50 mm × 2.1 mm, 1.8 um), with a ve- 86 locity of flow of 5 ul/s, with a sampling volume of 10 ul.” Please substitute the “u” with a proper “µ” symbol, the same for the concentrations in figure 1 ordinate/y axis.

Line 88 please substitute the “M” symbol, to indicate concentration, for mol/L or mol L-1. Additionally, “%” must be reserved for relative changes and not for concentrations please express as “mL/100 mL” or in “mL/L or mL L-1”. Analogously for line 97 “1 g/100 mL agarose”. Please revise the rest of the manuscript

Results

Lines 121-127 It is easier for a reader to follow if the mean concentrations for each mycotoxin/group are explicitly stated here.

Discussion

Lines 245-246 Please recheck the name of the last toxin. Also reference [12] does not mention anything about Alternaria toxins. Please add a more appropriate reference.

Line 246 “The abundance was also increased” The abundance of what? Please be a little more specific.

Line 246-247 Please provide a plausible explanation for this behavior.

While Alternaria seem to be prevalent as part of the microbiota of stored hay and capable of producing mycotoxins, why did none of Alternaria toxins included in the research?  

Author Response

Dear Editor and Reviewer

Thank you very much for evaluating our paper.

Thank you for your letter and for the reviewers’ comments concerning our manuscript entitled “Longitudinal investigation of the native grass hay from storage to market reveals mycotoxin associated fungi” (Microorganisms-1731131). We will be happy to edit the text further, based on helpful comments from editor and reviewers. We appreciate the editor very much for their positive and constructive comments and suggestions. According to the comments, this revised manuscript was checked by native speakers of English for editing English grammar. The comments and suggestions are not only helpful for us to revise and improve our manuscript, but also benefit our further research. We hope that our paper much better quality than before.

Best regards,

Dr. Shuai Du

E-mail: dushuai_nm@sina.com

Dr. Yushan Jia

E-mail: jys_nm@sina.com

Reviewer 2 Report

The objective of this study was to characterize the fungal diversity and mycotoxins (AFB1, DON, ZEA and OA) of native grass hay in different storage periods.
My main concerns are:
1. the abstract and materials do not explicitly state the number of samples tested. Under the graphs, the number is 6. Is it a total of 6 samples or 6 in each sampling?
2. The method of sampling is not described, which is not acceptable in such a study.
3. Where and how is the hay stored? Under what conditions? What about the weather conditions?
4. Why did you use the Chao1 index and the Shannon index?
5. The method for mycotoxin analysis is not sufficiently described.
6. Explanations of DON in ZEN concentrations and fungal diversity are missing. Which of the detected fungi can produce DON, ZEN and OA?

Some minor recommendations:
Line 62: add »some« before mycotoxins

Line 234: Ceniti and al – put the number instead of names

Author Response

(The authors gave the same response as above.)

Reviewer 3 Report

The manuscript presents a longitudinal investigation of the native grass hay in different storage periods. The theme is interesting and the topic covers the area which is neglected lately. Used references are adequate and support statements. The background is well described. Results are well presented.

Materials and methods have a lot of lack. It is not described how sampling was provided. How many samples of each storage period are sampled? Did the authors store samples before analyses? How did they prepare samples for mycotoxin analyses, and how for moulds DNA? What were LOD and LOQ of used method for mycotoxin analyses, and which standard was used?

In the results, it should be written obtained values of analyzed mycotoxins, not just be presented in graphs.

Also, in the results are mentioned tools that are not described in the Materials and methods section (Lines 146 and 182. Additionally, in Figure 4, suddenly is mentioned "Rumen fungal taxa" what is not mentioned never before and after in the text.

The same comment is for "multiple physicochemical analyses" (Line 201), which is not presented in this study. Also, macroscopically findings are in the discussion section (Line 223), that are not mentioned before. Those methods are not described in the material and methods section. In discussion section also is missing comments about ZEA and DON producers, which are not even detected in PCR amplification.

Author Response

(The authors gave the same response as above.)

Round 2

Reviewer 2 Report

Dear authors,

thank you for answering the questions and for the additional explanations in the text. However, I still have some comments for discussion. In my opinion, the links between the found mycotoxins and fungi are not sufficiently described. The sentence about DON, ZEA and Fusarium was added, but what about Chactomella?

Author Response

Dear Editor and Reviewer

Thank you very much for evaluating our paper.

Thank you for your letter and for the reviewers’ comments concerning our manuscript entitled “Longitudinal investigation of the native grass hay from storage to market reveals mycotoxin associated fungi” (Microorganisms-1731131). We will be happy to edit the text further, based on helpful comments from editor and reviewers. We appreciate the editor very much for their positive and constructive comments and suggestions. The comments and suggestions are not only helpful for us to revise and improve our manuscript, but also benefit our further research. We hope that our paper much better quality than before.

Best regards,

Dr. Shuai Du

E-mail: dushuai_nm@sina.com

Dr. Yushan Jia

E-mail: jys_nm@sina.com

Reviewer 3 Report

It can be seen that the authors improved the manuscript according to reviewers' comments. Still, some improvements should be made.

Methods are still problematic. It is well known that all methods from the article should be repeatable; meaning that after reading the description of the method everyone should be able to do it, with comparable results. After reading the presented method, a lot of vagueness is present. It is not mentioned what solvent was used for samples preparation for mycotoxin analyzes. Additionally, it is still not a precise performance of the used spectrometric method (LOD, LOQ, used standards). Numbered references are not enough to get a picture. Therefore, please be more specific when describing the used methods

Author Response

(The authors gave the same response as above.)
